# Clinical Characteristics and Post-Operative Outcomes in Children with Very Severe Obstructive Sleep Apnea

**DOI:** 10.3390/children9091396

**Published:** 2022-09-15

**Authors:** Nancy Saied, Roberto Noel Solis, Jamie Funamura, Joy Chen, Cathleen Lammers, Kiran Nandalike

**Affiliations:** 1Department of Anesthesiology and Pain Medicine, University of California, Davis, CA 95817, USA; 2Department of Otolaryngology-Head & Neck Surgery, University of California, Davis, CA 95817, USA; 3Division of Pediatric Pulmonology, University of California, Davis, CA 95817, USA

**Keywords:** adenotonsillectomy, polysomnogram (PSG), pediatric obstructive sleep apnea (pediatric OSA), sleep disordered breathing in children, tonsillar hypertrophy

## Abstract

Available information on clinical characteristics and post-operative outcomes in children with very severe obstructive sleep apnea (OSA) is limited. Our study evaluates the clinical features and polysomnographic (PSG) variables that predict post-operative outcomes in children with an obstructive apneal hypopnea index (AHI) of more than 25 events/hr. In this study from a single tertiary care center, we performed a retrospective chart review of patients with an AHI > 25/hr, who underwent tonsillectomy and adenoidectomy (T&A) between January 2016 and September 2021. In total, 50 children were included in the study: 26.0% (13/50) of children experienced post-operative respiratory events and four children needed intubation and ventilator support. Compared with children without respiratory events, children requiring post-operative respiratory interventions were younger (4.4 ± 5.2 vs. 8.0 ± 5.2 years; *p* = 0.04), had higher pre-operative AHI (73.6 ± 27.4 vs. 44.8 ± 24.9; *p* < 0.01), lower oxygen nadirs (70.0 ± 13.0% vs. 83.0 ± 7.0%; *p* < 0.01), and had lower body metabolic index Z-scores (−0.51 ± 2.1 vs. 0.66 ± 1.5; *p* < 0.04). Moderate to severe residual OSA was identified in 70% (24/34) of children with available post-operative PSG; younger children had better PSG outcomes. Our study shows that post-operative respiratory events are frequent in children with very severe OSA, particularly with an AHI > 40/h, younger children (<2 years of age), lower oxygen saturation (SpO_2_), and poor nutritional status, necessitating close monitoring.

## 1. Introduction

Sleep-disordered breathing (SDB) is a general term describing breathing difficulties occurring during sleep. Obstructive sleep apnea (OSA), one of the common forms of SDB, affects about 2–5% of the general pediatric population [1]. OSA is defined as recurrent episodes of partial or complete upper airway obstruction during sleep, causing disruption in normal sleep architecture and gas exchange during sleep. Childhood obesity, craniofacial anomalies, and neuromuscular disorders significantly increase the risks of OSA [2]. OSA in children is linked to poor behavioral and emotional health, metabolic alterations, pulmonary and systemic hypertension, and overall decreased quality of life [3,4,5,6,7]. Therefore, appropriate diagnosis and management should be established early, to decrease such morbidities.

Adenotonsillar hypertrophy is the most common reason for OSA in children, with tonsillectomy and adenoidectomy (T&A) considered the first line of treatment for pediatric OSA, with treatment success rates ranging from 60 to 80% in developmentally normal children [8,9]. The presence of severe or very severe OSA on polysomnography (PSG) is considered one of the strongest risk factors for residual OSA [9,10]. Moreover, the severity of OSA and gas exchange abnormalities have been observed as independent risk factors for perioperative respiratory complications, including hypoxemia and need for respiratory interventions [11,12,13,14,15]. Although severity of OSA is considered a risk factor for residual OSA and post-operative complications, there are no studies evaluating the clinical parameters and polysomnography findings that would predict the peri- and post-operative consequences in children with very severe OSA. In fact, there is no consensus for the definition of “very severe” OSA, with previous studies using apnea–hypopnea index (AHI) values ranging from ≥20 to 30 events/hr to diagnose very severe OSA [10,16].

This study aims to assess the rates of peri- and post-operative respiratory complications after T&A among children with very severe OSA and to identify clinical features and PSG findings that predict the risk for post-operative complications. As a secondary measure, we also compared the pre- and post-operative PSG findings in these children and analyzed the influence of clinical characteristics and post-operative complications on residual OSA. 

## 2. Materials and Methods

This study is a retrospective cohort study conducted at the University of California, Davis (UCD) between January 2016 and September 2021. Approval was obtained from the UCD Institutional Review Board (1681320-1) prior to data extraction from electronic medical records (EMR). Children 0–17 years old with very severe OSA (defined as an AHI > 25/hr), were identified from the PSG database (Cadwell Easy III PSG, Kennewick, WA, USA). Patients with very severe OSA via PSG, who underwent T&A with or without post-operative PSG, were included (n = 50). We also included patients who underwent supplementary procedures to address OSA, in addition to T&A, based on sleep endoscopy and surgeon’s clinical findings. Patients who underwent T&A prior to PSG or those with incomplete records were excluded. Surgical procedures were performed by a pediatric otolaryngologist at UCD. All patients were deidentified prior to data analysis.

For all subjects, data were collected on age, gender, height, weight, race, ethnicity, and presence of co-morbidities, such as asthma, isolated craniofacial conditions that are not part of a syndromic diagnosis (e.g., cleft palate, Pierre Robin sequence), syndromic diagnosis, neuromuscular disease, and prematurity. Data were also collected on the presence of pulmonary hypertension (PH), based on the echocardiogram diagnosis performed prior to surgical intervention. Body mass index (BMI) z-score was calculated using data charts from the World Health Organization for children aged 0 to 2 [17] and the Centers of Disease Control and Prevention growth charts for children 2 to 18 years [18]. Obesity was defined as a BMI z-score of more than 1.66.

At the time of the study, it was our children’s hospital’s practice to observe overnight, all children with an AHI ≥ 10/hr, with continuous pulse oximetry monitoring. Data were, therefore, collected on perioperative and post-operative respiratory events such as difficult intubation, opioid use, need for oxygen in the post-operative recovery unit, and post-operative respiratory interventions such as supplemental oxygen, high flow nasal cannula (HFNC) support and intubation and mechanical ventilator support. The patients were divided into two groups based on the presence or absence of the post-operative (overnight) respiratory interventions. Demographics and PSG parameters were compared between the two groups to identify risk factors for post-operative respiratory complications. Patients were further stratified by the level of post-operative respiratory interventions (oxygen vs. HFNC vs. intubation), and demographics and pre-operative sleep study characteristics were analyzed between the groups. Data were also collected on post-operative emergency room visits for the first three weeks after the surgery.

Data on PSG were collected from the PSG database (Cadwell Easy III PSG Washington, U.S.A.) from the sleep laboratory. Sleep staging, scoring of arousals, awakenings, apneic and hypopneic events were manually performed per previously published standard criteria [19]. The following features in each PSG were evaluated: total sleep time, sleep latency, sleep efficiency, proportion of time spent in different sleep stages including non-rapid eye movement (NREM) and rapid eye movement (REM), arousal index, apnea hypopnea index (AHI), REM AHI, central AHI, and end-tidal CO_2_ (ETCO_2_) or transcutaneous CO_2_ (tcpCO_2_). CO_2_ retention was defined as greater than or equal to 25% of sleep spent in ETCO_2_ or tcpCO_2_ more than 50 mmHg. AHI was defined as the total count of overnight respiratory events divided by the total sleep time in hours and was used to diagnosis OSA and assess its severity [19]. We used an AHI cut-off of 25 events/hr to define very severe OSA [10,16]. PSG studies with technical flaws or inadequate data were also excluded. If a patient had more than one test, then only the most recent diagnostic PSG prior to surgery (if applicable) was included.

All children with very severe OSA were recommended to undergo a repeat PSG to evaluate residual OSA three months after intervention. Complete resolution of OSA was defined as AHI < 1/hr, and partial resolution as AHI < 5/hr. Next, clinical characteristics and post-operative respiratory complications were compared between the groups who had complete resolution of OSA vs. partial resolution of OSA. Patients with significant residual OSA were continued to be followed by pediatric services and offered sleep endoscopy and second stage surgical procedures or continuous positive airway pressure therapy (CPAP) depending on the age group, co-morbidities, clinical presentations, and patient and/or family preference.

Statistical analysis: to identify a difference of at least 20 events/hr in a population in which approximately 25% were anticipated to have respiratory events requiring some form of intervention, a sample size of 40 was calculated, with a minimum of 30 patients in the non-respiratory event group and 10 in the respiratory event group (mean AHI 45 and 65 events/hr, alpha = 0.5, beta = 0.2). Continuous clinical and demographic characteristics were reported with mean with standard deviation, and median with interquartile range for normal and non-normal distributions, respectively. Univariate comparisons were made with Chi-square and Fisher exact (<5) tests for categorical variables, t-tests for normally distributed, and Mann–Whitney U tests for non-parametric continuous variables. Comparisons of post-operative interventions were made by ANOVA and Kruskal–Wallis analysis. All *p* values were the product of 2-sided tests with *p* < 0.05 considered statistically significant. All statistical analyses were conducted with IBM SPSS Statistics for Macintosh, Version 27.0 (IBM Corp, Armonk, NY, USA).

## 3. Results

### 3.1. Demographic and PSG Data

A total of 50 children with a mean age of 7.1 ± 5.4 years, diagnosed with very severe OSA were included in the study. The majority (64%) of included subjects were male and 40% were identified as Caucasian. A total of 38% of the children (19/50) were diagnosed with co-morbidities, and obesity was the most prevalent co-morbidity (28%). A total of 16 of 50 children underwent diagnostic echocardiogram to rule out PH due to their diagnosis of very severe OSA, and two of them were diagnosed with PH; both patients had underlying cardiac conditions. One patient was diagnosed with PH in the post-operative period. Mean pre-operative total AHI was 52.3 ± 28.3, with a SpO_2_ nadir of 79.9 ± 10.3%. A total of 19 (38%) of children also had concurrent sleep hypoventilation. Of the 50 children studied, six (12%) underwent another intervention, such as a supraglottoplasty, inferior turbinate reduction, and/or lingual tonsillectomy, concurrently with T&A. All patients were observed overnight and 20% required more than one night’s observation (range 2–14 days). While 8% of patients visited the emergency room or were re-admitted within a three-week post-operative period, only one was admitted for respiratory distress. 

### 3.2. Perioperative Data

In the perioperative period, 16 out of 50 patients (32%) experienced desaturation outside of the PACU (SpO_2_ < 92%), with 13 (81.3%) of those patients requiring overnight supplemental oxygen. Out of these 13 patients who needed supplemental oxygen, four required HFNC therapy and four needed intubations. Of note, three patients needed multiple respiratory interventions such as supplemental oxygen, HFNC therapy and eventually intubation (Table 1). 

Table 2 compares the characteristics of children with a documented overnight respiratory event post-surgery versus children without reported complication. Those requiring respiratory interventions were younger (mean age 4.4 ± 5.2 vs. 8.0 ± 5.2 years; *p* = 0.03) with higher pre-operative AHI (73.6 ± 27.4 vs. 44.8 ± 24.9; *p* < 0.021) and lower SpO_2_ nadir (70.0 ± 13.0% vs. 83.0 ± 7.0%; *p* < 0.01). Additionally, those with respiratory interventions had lower z-score weights at the time of surgery (−0.51 ± 2.1 vs. 0.66 ± 1.2; *p* < 0.04). In patients with a lack of post-operative respiratory events, the median length of stay was one day, whereas, for those with a post-operative respiratory event, the median length of stay was two days (*p* < 0.01). There were no significant differences in gender, race, co-morbidities and the type of intervention and other PSG parameters aside from AHI and SpO_2_ nadir, between the two groups. 

Patients were further stratified by their need for post-operative respiratory interventions, and pre-operative risk factors (age, BMI-z and sleep study characteristics, Table 3). In comparing the groups, patients who received respiratory support post-operatively had a significantly higher pre-operative AHI, O_2_ saturation nadir and lower BMI-z scores compared with those not needing airway interventions. Children < 2 years of age and AHI > 40 events/hr (8/13 children) appear to be at increased risk for post-operative respiratory interventions. Of the four children who ended up needing intubation in the post- operative period, all four were < 2 years of age, three out of four had underlying co-morbidity, all but one child was undernourished at the time of surgery, and all had an AHI > 50 events/hr and O_2_ saturation nadir of less than 70%. One patient, with Trisomy 21, remained electively intubated following the procedure, and the other three were re-intubated for respiratory distress in the post-operative period and remained intubated for a duration of 3–5 days. These three children, who remained intubated on day 2, were eventually diagnosed with viral and bacterial lower respiratory tract infections and all three were treated with antibiotics. One patient was also diagnosed and treated for PH in the post-operative period. 

Table 4 summarizes the changes in PSG parameters in children (34/50) with PSG studies both pre- and post-T&A. There were significant improvements in many sleep parameters following treatment, including: total sleep time in minutes, sleep efficiency percentage, REM duration in minutes, AHI, arousal index, and SpO_2_ nadir in percentage. There was a mean reduction of AHI from 46.4 to 13.7 (*p* < 0.01) and improvement in SpO_2_ nadir from 79.9% to 88.5% (*p* < 0.01). Of note, none had complete resolution (by AHI < 1 criteria) however, 10 out of 34 patients (29.4%) had partial resolution (AHI < 5) of OSA. Children with AHI < 5 on post-operative sleep study were significantly younger with a similar pre-operative total sleep time AHI of 49.7 ± 20.5 compared with 51.2 ± 27.7 (*p* = 0.957). There was a higher prevalence of co-morbid conditions in children with moderate to severe residual OSA (AHI ≥ 5 events/hr). Persistent moderate or severe OSA was not predicted by post-PACU desaturations, need for supplemental O_2_, HFNC/CPAP, or intubation (*p* = 0.68, 0.84, 0.83, 0.83, respectively).

## 4. Discussion

Our study shows that children younger than two years of age and children with severe OSA (AHI > 40) are at higher risk for post-operative respiratory interventions. Our study also shows that the children with lower SpO_2_ nadir and undernourished children are at increased risk for requiring post-operative respiratory interventions. The majority of our patients (~70%) had moderate to severe residual OSA, with younger children having higher rates of resolution on PSG, and older children and children with co-morbidities being at higher risk for residual OSA. The presence of post-operative respiratory events did not predict residual OSA in our cohort. 

There are limited data on the incidence and risk factors for adverse post-operative respiratory complications in children with very severe OSA. Studies in children with severe OSA (defined as AHI > 10 events/hr) have reported an overall respiratory adverse event of 14.8–26.8% [12,13,20] and significant correlation between respiratory complications and AHI as well as oxygen saturation nadir. Similar to our findings, prior studies have reported that children requiring post operative respiratory support such as oxygen of more than 2 L/min or intubation, were younger with a mean AHI > 40 and oxygen nadir of <74% [12,14,20]. Based on these findings, it appears that children under two years of age with AHI > 40 and significant oxygen desaturations, may benefit the most from being monitored in the intensive care setting post-operatively. In a recent study evaluating children with extreme OSA (AHI > 100), the authors reported that 11 out of 28 (39.3%) children required respiratory support in the post-operative period; they identified longer duration with SpO_2_ < 90% and lower SpO_2_ nadir on pre-operative PSG as predictive factors for respiratory events, while AHI was not predictive [21]. Similarly, work by Lim et al. reported time with SpO_2_ < 90% to be the strongest predictor of post operative respiratory events in their cohort of children with severe OSA [12]. Even though we have not collected these data for our study cohort, we have noted that patients requiring intubation had significant gas exchange abnormalities with extended time spent with saturation less than 90%. It could be that the duration of time spent in saturation less than 90% is a better predictor of adverse events than the single O_2_ nadir value. Future studies should include this parameter in analyzing the risk factors for post-operative complications. 

It is interesting to note that children who experienced post-operative respiratory complications had a lower body weight, as represented by z-score, compared with those who did not. Even though obesity was the most common co-morbidity in our cohort, post-operative respiratory complications were seen more commonly in undernourished children than the obese children. Previous studies have demonstrated a correlation between underweight status and OSA [22,23]. It is thought that OSA leads to decreased growth, as well as increased energy expenditure during sleep, contributing to weight loss. This is further supported by data demonstrating that body weight normalizes following T&A [24]. Thus, in our study, lower body weight may have reflected more severe pre-operative OSA, which placed these patients at higher risk for post-operative complications. In addition, malnutrition alone has been independently associated with poor surgical outcomes, wound healing, and length of stay. It is essential that pre-operative evaluation integrates nutritional evaluation for risk stratification and nutritional optimization prior to T&A. 

Of note, pulmonary hypertension (PH) was noted in two of 16 of our patients pre-operatively, and both children had underlying cardiac conditions. The third patient, a patient with achondroplasia and chronic lung disease, was diagnosed with PH only in the post operative period when his respiratory distress worsened, needing intubation for several days post-operatively. Despite the relationship between OSA and pulmonary hypertension, the literature regarding this in children is limited. A recent study by Maloney et al. has identified CO_2_ retention on the overnight PSG to be the significant risk factor for PH in children with severe OSA (AHI > 10/hr) [6]. Since our cohort included children with very severe OSA and gas exchange abnormalities, one would expect to see a higher incidence of PH. Although we now routinely screen patients with very severe OSA for PH, that was not the case at the time of data collection, and only 16 of 50 children underwent diagnostic echocardiogram to rule out PH due to their diagnosis of very severe OSA. Prior studies have shown that surgery or non-invasive ventilation will significantly improve elevated pulmonary pressures in a considerable number of children [25]. Further research needs to be undertaken to better understand the true concurrence of PH in children with very severe OSA, with more definitive diagnostic measures than echocardiogram, that would help address the post operative complications in these high-risk children. 

When we further explored the clinical course of children who needed the highest level of respiratory intervention, intubation, and mechanical ventilation (n = 4), we noted that three of these children were also diagnosed and treated for bacterial lower respiratory tract infections. Although the role of antibiotics is controversial in children who undergo T&A, prophylactic antibiotics have been shown to decrease post-operative complications in some children with OSA [26]. In addition to young age and poor nutritional status, adverse respiratory events could also be linked to the concomitant respiratory infections with a potential role of antibiotic therapy [27]. These findings show the importance of timely pre-operative screening and appropriate medical planning for these high-risk children awaiting an urgent surgical intervention. Children < 2 years of age with severe OSA and gas exchange abnormalities as well as children with nutritional failure need to be on higher alert for post-operative respiratory monitoring. Screening for respiratory pathogens and selective use of antibiotics can be considered in this high-risk population to minimize the risk of respiratory interventions post-operatively.

While not the primary aim of our study, we compared pre-operative and post-operative PSG results of the 34 of 50 (68% of) patients who underwent post-operative PSG. Pediatric OSA success rates following T&A range from 60 to 80% in developmentally normal children [8,9]. However, in a study of 74 children with very severe OSA (AHI > 30 events/hr), who had both pre-operative and post-operative PSGs, 80% had partial resolution and only 32% had complete resolution of OSA [10]. It is important to note that this study excluded patients with craniofacial and genetic disorders, which were included in our study. Only a third of the patients in our study had partial resolution (AHI < 5 events/hr) and none had complete resolution (AHI < 1 events/hr). The older age group and presence of co-morbid conditions predicted the risks of residual OSA. The change in AHI after surgery was not predicted by respiratory events in the perioperative period. Residual OSA is treated with sleep endoscopy and second stage surgical procedures, continuous positive airway pressure therapy, weight reduction or medications, depending on the clinical presentation [1]. Tailoring treatment options with sleep endoscopy and customized surgical approach might sound beneficial to minimize the risks of residual OSA in these children. However, in our cohort, sleep endoscopy and customized therapies were planned for some children but were aborted due to their instability in the operative period, especially in younger children. Adenotonsillectomy should still be offered as a first line therapy in younger children, but a more customized approach can be attempted in older children as a first line therapy. Children with co-morbid conditions need to be aware of the high risk of residual OSA and emphasis must be placed on nutritional counseling/weight loss prior to surgery [28]. 

The limitations of this study include many of those which are inherent to retrospective studies. We relied on accurate documentation within the EMR to characterize respiratory events during patient hospitalization and in the post-operative setting. It is possible that patients living far from our center could have been admitted for post-operative respiratory events after discharge, which were not captured in the records. Although it is an institutional recommendation to undergo a post-operative PSG after intervention in patients with severe OSA, only 68% of patients had PSGs to analyze. There may have been selection bias, as parents who were satisfied with the outcome of surgery could have decided not to pursue the recommendation of a repeat PSG, therefore, our rates of resolution could be lower than reported. In addition, the cohort of 50 patients was a relatively small size and prevented us from running any sub-analysis, such as analyzing outcomes in children with and without co-morbidities, separately. Lastly, there were a few children (n = 6) who underwent multiple procedures for severe OSA. Even though we did not observe a higher incidence of post-operative events in these children, future large-scale studies are needed to further explore these confounding factors. 

## 5. Conclusions

In summary, our study suggests that younger children (particularly age < 2 years) with greater severity of OSA (AHI > 40 and lower SpO_2_ nadir), and undernourished children are at higher risk for post-operative respiratory complications needing airway interventions. Families and care providers need to be educated about these risk factors prior to interventions and children with these risk factors should be monitored in a higher care setting post adenotonsillectomy. The children with very severe OSA should undergo a more rigorous pre-operative screening that includes nutritional assessment, evaluation of co-morbidities such as PH, and screening for respiratory infections. Timely assessment and management of these risk factors may mitigate some of the post-operative complications in these children. Our cohort showed that the older children and children with co-morbidities are at higher risk for residual OSA, and these children and families need to be educated about this risk prior to interventions.

## Figures and Tables

**Table 1 children-09-01396-t001:** Adverse perioperative respiratory events of the cohort (n = 50).

Events: No. (%)	Overall Group
Difficult intubation	4 (8.0)
Post-operative desaturation	16 (32.0)
Supplemental oxygen	13 (26.0)
HFNC/CPAP administration	4 (8.0)
Intubation	4 (8.0)
ED visit/re-admission ^	1 (2.0)

Abbreviations: ED: emergency department; HFNC: high flow nasal cannula; CPAP: continuous positive airway pressure, ^ Re-admitted for respiratory issues. Other reasons for ED visits or re-admissions not included.

**Table 2 children-09-01396-t002:** Univariate analysis of baseline characteristics among children with pre-operative AHI ≥ 25.

	No. Post-Operative Respiratory Events N = 37	Presence of Post-Operative Respiratory Events N = 13	*p* Value
Age in years, mean (SD)	8.0 (5.2)	4.4 (5.2)	0.04 ^t^
Sex, No. (%)			
Female	13 (35.1)	5 (38.5)	1.00 ^c^
Male	24 (64.9)	8 (61.5)	
Race/Ethnicity, No. (%)			0.66 ^f^
Asian	5 (13.5)	2 (15.4)	
African-American	7 (18.9)	3 (23.1)	
Caucasian	15 (40.5)	5 (38.5)	
Hispanic	8 (21.6)	1 (7.7)	
Other	2 (5.4)	2 (15.4)	
Co-morbidities, No. (%)	14 (37.8)	5 (38.4)	0.97 ^c^
Asthma	8 (21.6)	2 (15.4)	0.71 ^f^
Craniofacial condition	6 (16.2)	4 (30.8)	0.42 ^f^
Syndromic diagnosis	6 (16.2)	4 (30.8)	0.42 ^f^
Neuromuscular disease	0 (0)	0 (0)	
Pulmonary HTN *	1 (2.6)	1 (7.7)	0.16 ^f^
Obesity	12 (32.4)	2 (15.4)	0.30 ^f^
Prematurity	3 (8.1)	3 (23.1)	0.17 ^f^
Z-score weight, mean (SD)	0.66 (1.52)	−0.51 (2.07)	0.04 ^t^
Intervention (%)			0.64 ^f^
T&A alone	33 (89.2)	11 (84.6)	
T&A with other intervention ^	4 (10.5)	2 (15.4)	
Opioid administration post-PACU Pre-operative sleep study data, mean (SD)	2 (5.2)	0 (0)	1.00 ^f^
Total sleep time, min	351.6 (87.2)	328.2 (124.0)	0.46 ^t^
Sleep efficiency, %	79.2 (16.2)	76.3 (17.3)	0.36 ^t^
REM duration, min	45.7 (34.0)	49.4 (46.7)	0.96 ^t^
N1 Stage duration, min	30.6 (48.6)	36.7 (50.8)	0.69 ^t^
N2 Stage duration, min	156.5 (71.8)	152.6 (70.2)	0.74 ^t^
N3 Stage duration, min	107.9 (68.9)	89.5 (58.3)	0.28 ^t^
Arousal index	31.2 (24.8)	44.2 (33.0)	0.07 ^t^
AHI, events/hr	44.8 (24.9)	73.6 (27.4)	<0.01 ^t^
SpO_2_ nadir, %	83.0 (7.0)	70.0 (13.0)	<0.01 ^t^
Hypoventilation, No. (%) °	15 (46.8)	4 (66.7)	0.02 ^f^
Length of stay >1 day, No. (%)	0	10 (76.9)	<0.01 ^m^
Length of stay median, (IQR)	1 (1–1)	2 (2–3)	<0.01 ^m^
Opioid administration post-PACU	2	0	1.00 ^f^

Abbreviations: AHI: apnea–hypopnea index; HTN: hypertension; T&A: tonsillectomy and adenoidectomy; ^ supraglottoplasty, inferior turbinate reduction and/or lingual tonsillectomy; ^c^ Chi-squared test; ^f^ Fisher exact test; ^t^
*t*-test; ^m^ Mann–Whitney U test; * Echocardiogram to r/o PH was performed in only 16 of 50 patients prior to the procedure and one patient underwent echo post procedure; ° hypoventilation data for only 38/50 patients.

**Table 3 children-09-01396-t003:** Patient and pre-operative sleep study characteristics by occurrence of post-operative respiratory interventions, stratified by post-operative respiratory interventions.

	No. Post-PACU Desaturations N = 37	Highest Level of Respiratory Intervention Post-PACU			*p*-Value
		Supplemental O_2_ N = 8	HFNC/CPAP N = 1	Intubation N = 4	
Age in years, mean (SD)	8.0 (5.2)	5.9 (6.3)	3.2	1.8 (0.5)	0.12 ^a^
Z-score weight mean (SD)	0.66 (1.52)	0.27 (1.94)	−1.70	−1.79 (2.00)	0.02 ^a^
AHI, mean (SD)	41.0 (24.4)	46.2 (27.1)	94.2	85.2 (33.1)	<0.01 ^a^
O_2_ nadir %, (SD)	83.0 (7.0)	73.0 (16.0)	70.0	65.0 (3.0)	<0.01 ^a^
Length of stay, median (IQR)	1 (1–1)	2 (1–2.5)	2 (2–2)	4 (3–9.5)	<0.01 ^k^

^a^: ANOVA. ^k^: Kruskal–Wallis.

**Table 4 children-09-01396-t004:** Changes in PSG parameters for children with severe OSA following T&A (n = 34) ^.

Parameter	Pre-T&A	Post-T&A	Change (95% CI)	*p* Value ^t^
Total sleep time, min	345.5	366.6	+21.1 (−3.6 to 45.8)	0.58
Sleep efficiency, %	80.0	90.0	+10.0 (5.8 to 14.2)	<0.01
REM duration, min	48.9	71.0	+22.1 (10.5 to 33.7)	0.01
Stage N1 duration, min	32.1	22.3	−9.8 (−16.7 to −2.9)	0.37
Stage N2 duration, min	158.2	164.5	+6.3 (−9.4 to 22.0)	0.20
Stage N3 duration, min	106.4	117.6	+11.2 (−5.5 to 27.9)	0.47
Arousal index	32.7	14.1	−18.6 (−21.9 to −15.3)	<0.01
AHI, events/hr	46.4	13.7	−32.7 (−38.3 to −26.8)	<0.01
SpO_2_ nadir, %	79.9	88.5	+8.6 (7.4 to 10.1)	<0.01
Hypoventilation, no. (%) ^	19 (50.0)	13 (52.0)		0.57 ^c^

^ PSG parameters reported as mean values; ^t^
*t* test; ^c^ Chi-squared test. Abbreviations: AHI: apnea–hypopnea index; REM: rapid eye movement.

## Data Availability

Data is available and can be requested from Nancy Saied.

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
