# Peer review of "Clinical Characteristics and Post-Operative Outcomes in Children with Very Severe Obstructive Sleep Apnea"

_children, 2022, doi:10.3390/children9091396_

Round 1
Reviewer 1 Report
The paper "Clinical Characteristics and Post-operative Outcomes in Children with Very Severe Obstructive Sleep Apnea" by Nancy Saied et al. is a retrospective single center analysis of a 50 children with surgery procedures. There are small issues that can be solved.
Line 66. Did you have children with multiple surgical procedures?
Line 76, echocardiogram diagnosis of PHTN- a limitation that you can discuss and add at the chapter.
Line 79. Reference 16 needs editing.
Line 88. Did you have video- assisted PSG?
Line 96. Reference 16...Manually validated PSG?
Line 124. "Mean pre-operative total AHI was 52.3± 28.3, with a SpO2 nadir of 79.9±10.3%. (38%) of children also had concurrent sleep hypoventilation." How many were with obesity- hypoventilation syndrome?
Line 190. Despite interventions, the AHI is still severe (13.7). Can you comment more about this situation? Did they used additional CPAP?
Author Response
Reviewer 1:
The paper "Clinical Characteristics and Post-operative Outcomes in Children with Very Severe Obstructive Sleep Apnea" by Nancy Saied et al. is a retrospective single center analysis of a 50 children with surgery procedures. There are small issues that can be solved.
We thank the reviewer and have addressed the following concerns.
Line 66. Did you have children with multiple surgical procedures?
We did have 6 children (Table 2), 4 (10.5% ) without post operative complications and 2 (15.4) with post operative complications who underwent multiple procedures such as supraglottoplasty, inferior turbinate reduction and/or lingual tonsillectomy. This information is included in the results section as well as Table 2.
Line 76, echocardiogram diagnosis of PHTN- a limitation that you can discuss and add at the chapter.
We agree with the reviewer in that the ECHO is not the diagnostic test for PHTN and we have now included this information in discussion.
Line 79. Reference 16 needs editing.
Thank you, we have now updated this reference.
Line 88. Did you have video- assisted PSG?
Thank you for asking to clarify. We have used Cadwell Easy III PSG, 32 channel, video assisted PSG system and updated this information in the manuscript.
Line 96. Reference 16...Manually validated PSG?
Thank you once again for asking to clarify. The studies were all manually scored per the standard reference criteria (Iber C, A.-I.S., Chesson Jr. AL, Quan SF. The AASM Manual for the Scoring of Sleep and Associated Events: Rules, Terminology and Technical Specifications. 1st ed. Westchester, Illinois: American Academy of Sleep Medicine; 2007). We have now updated the reference.
Line 124. "Mean pre-operative total AHI was 52.3± 28.3, with a SpO2 nadir of 79.9±10.3%. (38%) of children also had concurrent sleep hypoventilation." How many were with obesity- hypoventilation syndrome?
Thank you for asking to clarify. Hypoventilation data was available only for 38/50 patients. This information is available in table 2. There were a total of 15 patients with obstructive hypoventilation, of which five patients had underlying obesity.
Line 190. Despite interventions, the AHI is still severe (13.7). Can you comment more about this situation? Did they used additional CPAP?
We have not systematically studied the long-term management or outcomes for these patients, but we routinely offer second stage surgical procedures or CPAP therapy for patients with significant residual OSA. The decision for second stage procedures or PAP therapy is driven by multiple factors including the age, co-morbidities, clinical presentations, and patient and/or family preference. We have now included this information in methodology.
Reviewer 2 Report
This reviewer appreciates the opportunity to evaluate this article with an interesting topic.
This retrospective work is about possible rates of peri- and post-operative respiratory complications after T&A among children with very severe OSA.
The manuscript had an apparent valid method, but I have some doubts. For example, in materials and methods, authors reported that some patients underwent multiple surgical procedures. Maybe this is a possible confounding factor for results interpretation?
Moreover, they also enrolled children affected by severe OSA and genetic syndromes. Maybe those conditions as a confounding factor?
Maybe it should be better to perform a more selected enrolment of the patients analyzing only the effective OSA effect on peri- and post-operative respiratory complications avoiding syndromic patients or multiple surgically treated.
Finally, since the sample is poorer and suitable for a preliminary study, I suggest further investigation to confirm the results.
Author Response
Reviewer 2:
This reviewer appreciates the opportunity to evaluate this article with an interesting topic. This retrospective work is about possible rates of peri- and post-operative respiratory complications after T&A among children with very severe OSA.
We thank the reviewer for taking time to review our article.
The manuscript had an apparent valid method, but I have some doubts. For example, in materials and methods, authors reported that some patients underwent multiple surgical procedures. Maybe this is a possible confounding factor for results interpretation?
We thank the reviewer for bringing up this valid point. We did have 6 children (Table 2), 4 (10.5%) without post operative complications and 2 (15.4) with post operative complications who underwent multiple procedures such as supraglottoplasty, inferior turbinate reduction and/or lingual tonsillectomy. This information is included in the results section as well as Table 1. The age group of the children ranged from 1.5 to 17.1 years and only 2 children were under the age four years. Since this is a small number, and since there were no differences in post operative events between the children who underwent T&A alone and multiple procedures, we believe that this may not be a confounding factor in interpretation of results.
Moreover, they also enrolled children affected by severe OSA and genetic syndromes. Maybe those conditions as a confounding factor? Maybe it should be better to perform a more selected enrolment of the patients analyzing only the effective OSA effect on peri- and post-operative respiratory complications avoiding syndromic patients or multiple surgically treated.
We thank the reviewers for bringing up this valid point. Even though it is ideal to include the more selected patients without co-morbid conditions, our study did not notice any significant difference between the children with and without co-morbid conditions in terms of post operative complications (14 (37.8%) vs. 5 (38.4%)). Our study indicates that younger children and children with AHI>40 events/hour. and undernourished children are at particularly at higher risk for post operative complications.
Finally, since the sample is poorer and suitable for a preliminary study, I suggest further investigation to confirm the results.
We do agree with the reviewer that the sample size is small, and we have included this as our limitation. Even though our sample size is small, our findings are significant. We agree that the future large-scale studies will be helpful to further explore the risks factors in children with very severe OSA. The larger studies should give an opportunity for sub-analysis of the patients with co-morbid conditions as well as patients undergoing multiple surgeries.
We would like to bring the reviewer’s attention to a recently published study by Lim et al. (Lim J, Garigipati P, Liu K, Johnson RF, Liu C. Risk Factors for Post-Tonsillectomy Respiratory Events in Children With Severe Obstructive Sleep Apnea. Laryngoscope. 2022 Aug 6. doi: 10.1002/lary.30317.) who assessed the risk factors for adverse post-operative outcomes in a larger group of children with severe OSA (N=887) and reported 14.8% adverse events in their patient population. Similar to our study, they report that the post operative adverse events were higher in younger children, children with a mean AHI>40 and oxygen nadir of <74%. They also report black race, neurological co-morbidities and % time <90% as risk factors. When they excluded the children with co-morbidities, younger age and % time <90% stood out to be strong risk factors. We have now referenced this paper and strengthened our discussion as well.
Reviewer 3 Report
For young children (under 2 years), full adenotonsillectomy is not suggested for immunity concern. Further, laryngomalacia may play more role in these children and consequently they needed to be separated into another group instead of routine T&A group in terms of risk factors analysis in adverse respiratory events.
Author Response
For young children (under 2 years), full adenotonsillectomy is not suggested for immunity concern. Further, laryngomalacia may play more role in these children and consequently they needed to be separated into another group instead of routine T&A group in terms of risk factors analysis in adverse respiratory events.
We thank the reviewer for providing this valuable feedback. We agree that that the tonsils and adenoid are the first line defense mechanisms, and it is not ideal to undergo adenotonsillectomy under age two. However, these patients underwent adenotonsillectomy based on the clinical findings of adenotonsillar enlargement in the setting of very severe OSA. There were 7/50 children under age two years and one of them had co-existing laryngomalacia and underwent supraglottoplasty in addition to adenotonsillectomy. Since there was only one patient with this diagnosis, we do not believe this patient needs to be excluded or separated into a different group.
Reviewer 4 Report
This manuscript is a retrospective chart review from a single tertiary care center tonsillectomy and adenoidectomy in very severe OSA.
It is an interesting and well-written paper.
I recommend its publication.
However, it needs minor improvements:
1.- Mat & Method
Sample size calculations is missing
Statistical Analysis Plan is missing
2.- During discussion
Main result in lines 202-210:
“Our study shows that the children younger than two years of age and children with greater severity of OSA (AHI>40) are at highest risk for post-operative respiratory interventions. Our study also shows that the children with lower Spo2 nadir and undernourished children are at higher risk for post-operative respiratory complications."
Discuss cut-off AHI>40 to define very severe OSA in children
Please cite:
Lim J, Garigipati P, Liu K, Johnson RF, Liu C. Risk Factors for Post-Tonsillectomy Respiratory Events in Children With Severe Obstructive Sleep Apnea. Laryngoscope. 2022 Aug 6. doi: 10.1002/lary.30317.
Billings KR, Somani SN, Lavin J, Bhushan B. Polysomnography variables associated with postoperative respiratory issues in children <3 Years of age undergoing adenotonsillectomy for obstructive sleep apnea. Int J Pediatr Otorhinolaryngol. 2020 Oct;137:110215. doi: 10.1016/j.ijporl.2020.110215.
Keamy DG, Chhabra KR, Hartnick CJ. Predictors of complications following adenotonsillectomy in children with severe obstructive sleep apnea. Int J Pediatr Otorhinolaryngol. 2015 Nov;79(11):1838-41. doi: 10.1016/j.ijporl.2015.08.021.
Discuss also %sleep time with O2 < 90% in your series
Author Response
Reviewer 4:
This manuscript is a retrospective chart review from a single tertiary care center tonsillectomy and adenoidectomy in very severe OSA. It is an interesting and well-written paper. I recommend its publication. However, it needs minor improvements:
We sincerely thank the reviewer for their review and valuable comments.
1.- Mat & Method
Sample size calculations is missing, and Statistical Analysis Plan is missing
We thank the reviewer for bringing this to our attention. We have now addressed this issue and addended the methodology in the revised paper.
2.- During discussion
“Our study shows that the children younger than two years of age and children with greater severity of OSA (AHI>40) are at highest risk for post-operative respiratory interventions. Our study also shows that the children with lower Spo2 nadir and undernourished children are at higher risk for post-operative respiratory complications." Discuss cut-off AHI>40 to define very severe OSA in children.
We agree with this suggestion and have now included this point in our discussion.
Please cite:
Lim J, Garigipati P, Liu K, Johnson RF, Liu C. Risk Factors for Post-Tonsillectomy Respiratory Events in Children With Severe Obstructive Sleep Apnea. Laryngoscope. 2022 Aug 6. doi: 10.1002/lary.30317.
Billings KR, Somani SN, Lavin J, Bhushan B. Polysomnography variables associated with postoperative respiratory issues in children <3 Years of age undergoing adenotonsillectomy for obstructive sleep apnea. Int J Pediatr Otorhinolaryngol. 2020 Oct;137:110215. doi: 10.1016/j.ijporl.2020.110215.
Keamy DG, Chhabra KR, Hartnick CJ. Predictors of complications following adenotonsillectomy in children with severe obstructive sleep apnea. Int J Pediatr Otorhinolaryngol. 2015 Nov;79(11):1838-41. doi: 10.1016/j.ijporl.2015.08.021.
We thank the reviewer for these great references, and we have now included the above and updated our references. We have strengthened our discussion by using these references including the most recent reference that was published after our submission of the first version of the paper.
Discuss also %sleep time with O2 < 90% in your series
Thank you for bringing up this point. This is a valuable point, but we do not have access to this data in our sleep reports and we could not include this in our data. We have now included this in our discussion.